# An Ultra-Low-Power Area-Efficient Non-Coherent Binary Phase-Shift Keying Demodulator for Implantable Biomedical Microsystems

**Milad Ghazi [1], Mohammad Hossein Maghami [1],\* , Parviz Amiri [1] and Sotoudeh Hamedi-Hagh [2],\***

[1] Faculty of Electrical Engineering, Shahid Rajaee Teacher Training University, Tehran 1678815811, Iran; milad.ghazi.sru@gmail.com (M.G.); pamiri@sru.ac.ir (P.A.)
[2] Department of Electrical Engineering, San Jose State University, San Jose, CA 95192, USA
\* Correspondence: mhmaghami@sru.ac.ir (M.H.M.); sotoudeh.hamedi-hagh@sjsu.edu (S.H.-H.)

**Abstract:** A novel non-coherent, low-power, area-efficient binary phase-shift keying demodulator for wireless implantable biomedical microsystems is proposed. The received data and synchronized clock signal are detected using a delayed digitized format of the input signal. The proposed technique does not require any kind of oscillator circuit, and due to the synchronization of all circuit signals, the proposed demodulator can work in a wide range of biomedical data telemetry common frequencies in different process/temperature corners. The presented circuit has been designed and post-layout-simulated in a standard 0.18 μm CMOS technology and occupies $17 \times 27$ μm$^2$ of active area. Post-layout simulation results indicate that with a 1.8 V power supply, power consumption of the designed circuit is 8.5 μW at a data rate of 20 Mbps. The presented demodulation scheme was also implemented on a proof-of-concept circuit board for verifying its functionality.

**Keywords:** implantable biomedical microsystems; data telemetry; low power; high data rate; binary phase-shift keying demodulation

---

## 1. Introduction

In Implantable Biomedical Microsystems (IBM), a wireless interface is used for transmission of power and data between the internal and external parts of the system (Figure 1). The most important issues in a wireless link for IBM are data transfer rate, power consumption, and chip area [1–3]. It is evident that in a forward data transfer, the maximum data rate is desired, especially in applications such as visual prostheses in which the implanted microsystem is in direct contact with the central neural system [4]. Furthermore, due to the power loss in the power-transmission circuitry and power dissipation in the tissue [5], the maximum frequency of the carrier signal for the IBMs is limited to a few tens of megahertz [1–5].

There have been proposed various modulation techniques for wireless data transfer [1,4,6]. Making comparisons between the various digital modulation schemes from different point of views, reveals that binary phase-shift keying (BPSK) structure is more appropriate in most IBMs [7–13]. Generally, this scheme provides proper insensitivity to amplitude noise, high data rate, and good power transfer efficiency. Moreover, if a capacitive link is utilized for data transmission, fast phase variations can be attained within every carrier cycle and data-rate-to-carrier-frequency (DRCF) ratios of as high as 100% are achievable in this scheme [1].

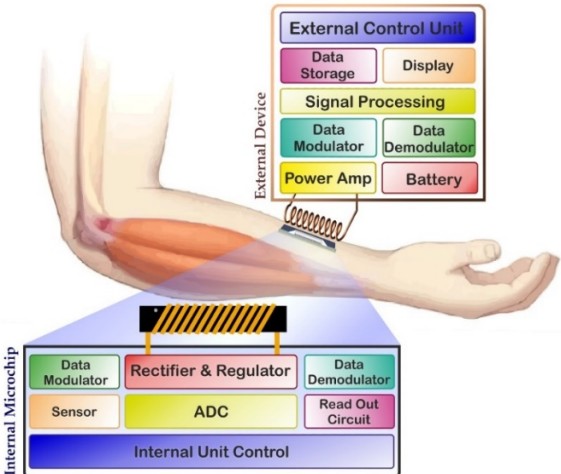

**Figure 1.** Simplified general block diagram of an Implantable Biomedical Microsystem (IBM).

BPSK demodulation can be realized in two ways: coherent and non-coherent structures. In coherent detection, phase synchronization between the received signal and the receiver is essential [8]. Most of the digital coherent demodulators consume lots of power and suffer from circuit complexity. Consequently, non-coherent BPSK demodulators are widely used in radio frequency applications and IBMs due to their lower circuit complexity, occupying less chip area, and lower power consumption, despite higher bit error rates [9,10].

Most of the non-coherent BPSK demodulation schemes are designed based on edge-detection technique [1,14], pulse width measurement [15], and filtering techniques [16]. Among all the benefits of these circuits, there are some disadvantages, like high power consumption and occupying a large area. This article introduces a new high-data-rate, low-power, non-coherent BPSK demodulator for fixed-in position implantable biomedical applications such as cochlear implants and visual prostheses, where the link properties such as distance between receiver and transmitter, orientation, and alignment are easily available [1]. The proposed demodulator is based on using a delayed digitized format of the input signal for the detection of the received data as well as for recovering the clock signal. The operation of the proposed circuit is closely comparable in terms of power consumption, occupied active area, and reliability to most of the existing similar works.

## 2. Proposed BPSK Demodulator and Clock Recovery Circuit

Figures 2 and 3 depict the block diagram of the designed demodulator as well as its essential theoretical waveforms, respectively. It should be noted that placing an automatic gain control as the first stage of the receiver is needed in applications where orientation and matching between the transmitting and receiving blocks are lost [17]. However, please note that as with fixed-in position implantable biomedical applications [1] where the link properties such as distance between receiver and transmitter, orientation, and alignment are easily available, it is not needed to use automatic gain control in the receiver. In these cases, the minimum required level of received signal for correct operation of the circuit is provided by the power amplifier in the external part according to the link properties. Moreover, in these cases, as the received BPSK signal level is high enough, power can also easily deliver wirelessly to the implant through the energy contained in the incoming BPSK data signal [18,19]. In this architecture, the rectified BPSK signal is digitized by logic inverters, to be used as an auxiliary clock with twice the BPSK frequency, and is called "2XClock". The digitized BPSK signal is then fed to the half-cycle delay block, containing a D flip-flop [20] which operates with 2XClock signal. For extracting data, a low-power data detector proposed in this work will retrieve original data from the digitized BPSK signal and a delayed digitized BPSK signal whose operation is described later

in this section. In addition, through comparing the digitized BPSK and data recovered signals, only by a low-power XOR logic gate [21], clock recovery will take place.

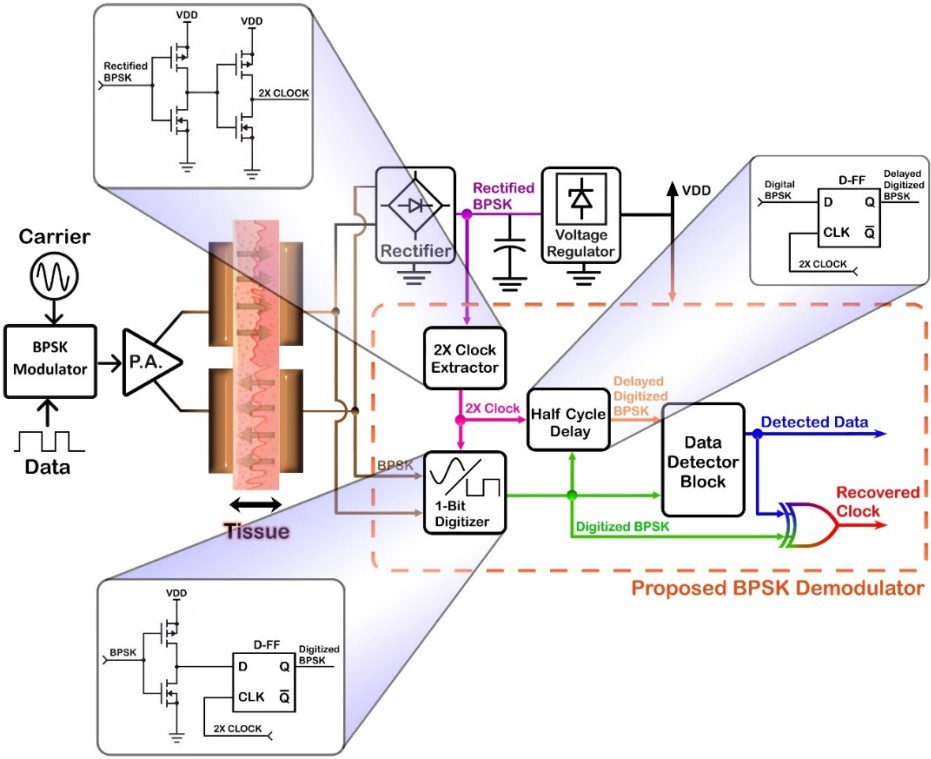

**Figure 2.** Block diagram of the presented binary phase-shift keying (BPSK) demodulator.

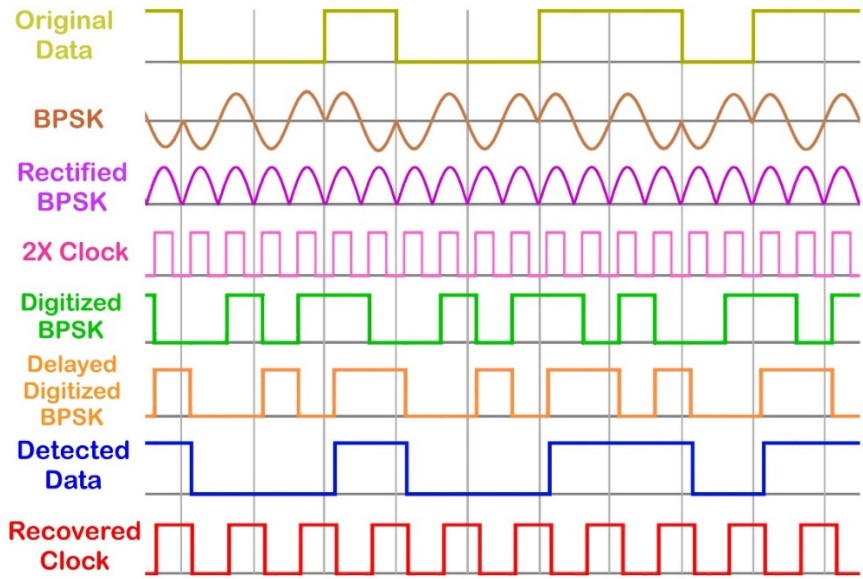

**Figure 3.** Key waveforms of the proposed demodulator.

It can be understood from Figure 3 that to retrieve the original data from the digitized BPSK and delayed digitized BPSK signals, a circuit is needed so that when "0, 0" ("digitized BPSK, delayed digitized BPSK," respectively) state is received, the output signal changes its value to "0" and remains unchanged while inputs are constant, and when "1, 1" state is received, the output signal changes its value to "1" and remains unchanged while inputs are not changed to "0, 0". It is important that in two another states ("0, 1" and "1, 0"), the output must not change and should retain its previous

value. As it is shown in Figure 3, some logical states between the digitized BPSK signal and delayed digitized BPSK signal never happen, for example, "0, 0" and "1, 1" after each other or "0, 1" after "0, 0" state. All situations that are possible to happen are illustrated in Figure 4.

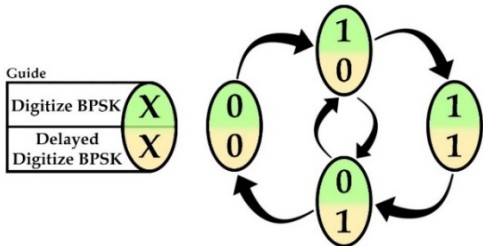

**Figure 4.** Possible states of data detector inputs.

According to above mentioned hints, a simple low-power circuit for retrieving data from existing signals is proposed (Figure 5). For understanding the operation of the proposed data detector, suppose that both digitized BPSK and delayed digitized BPSK signals are in low logic level. In this case, $M_{1,8,9}$ transistors are ON and $M_{3,4,6}$ devices are OFF. Therefore "B" node will be shortened to ground and accordingly, "A" node is connected to $V_{dd}$ via $M_{1,2}$ transistors. Likewise, $M_{10}$ will be ON and $M_5$ will be OFF. In this state, the output level is low, and it will remain unchanged. According to Figure 4, after "0, 0" state, only "1, 0" state can happen. In this case, again, $M_{1,2,10}$ Mosfets are ON and the output voltage retains its previous value. Now, when "1, 1" signals are received at data detector inputs, opposite the "0, 0" state, $M_{3,4}$ transistors are ON and "A" node will be shortened to ground, and then $M_7$ will be ON and given that the $M_6$ was ON, "B" node will be connected to $V_{dd}$. Consequently, $M_5$ will be ON and makes "A" node unchanged at the low level. Similarly, when inputs change to "1, 0" state, the output level will not change. In this way, the original data can be detected by the circuit proposed in Figure 5. Transistor dimensions of the data detector circuit are reported in Table 1.

**Table 1.** Transistor dimensions of the data detector circuit.

| Device Name | W/L |
|---|---|
| $M_{1,2,6,7}$ | 0.88 μm/0.18 μm |
| $M_{3,4,8,9}$ | 0.22 μm/0.18 μm |
| $M_{5,10}$ | 0.44 μm/0.18 μm |

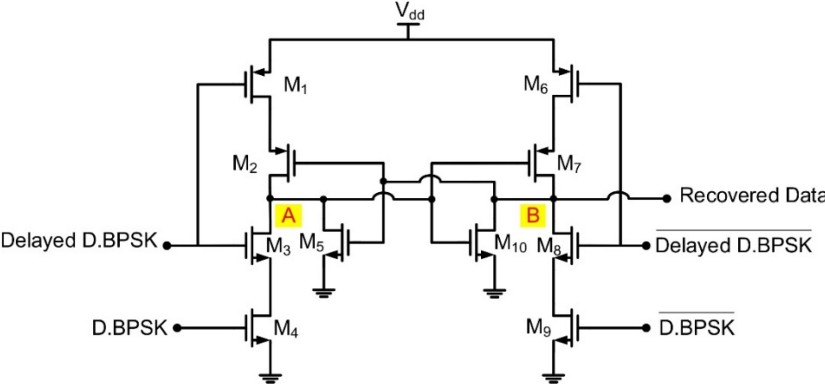

**Figure 5.** Proposed data detector.

## 3. Post-Layout Simulation and Experimental Results

The presented circuit has been designed and simulated in a 0.18 μm CMOS process. The designed modulator occupies a total area of $17 \times 27$ μm² (Figure 6). Important waveforms that resulted from

post-layout simulations in the typical NMOS and PMOS corner case at 20 MHz carrier frequency are shown in Figure 7. All traces are named with the ones used in Figure 2. A power-on-reset signal makes the reset signal needed to initialize the flip-flops and all other circuits at the beginning and makes them ready for the correct operation. According to post-layout simulations, demodulated data has a latency of 5 ns from the input BPSK signal which can be ignored in noncoherent detection.

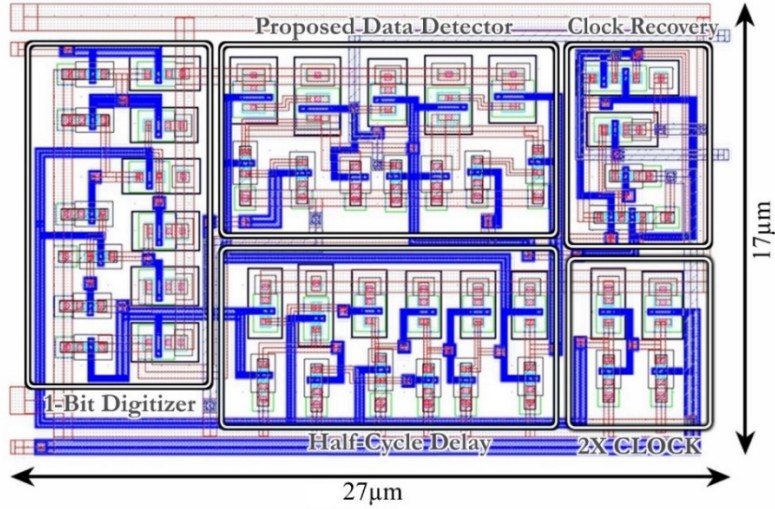

**Figure 6.** Layout view of the presented circuit.

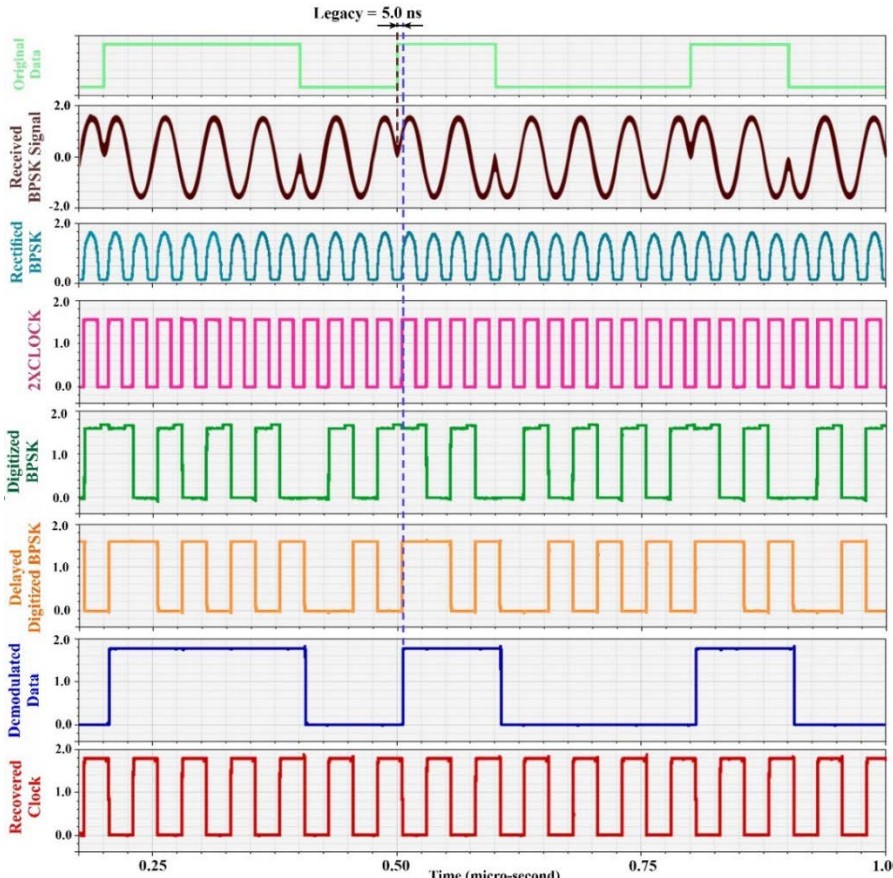

**Figure 7.** Transient response of the presented BPSK demodulator at 20 MHz carrier frequency.

The proposed circuit was post-layout-simulated in various process/temperature corners, with ±10% changes in power supply, for all of which correct operation was seen. The demodulated data and the recovered clock, as well as some other waveforms, in some process/temperature corners are shown in Figures 8–10 at a 20 MHz carrier frequency that resulted from post-layout simulations. The results of post-layout simulations for different process corners and temperature variations in 0.18 μm CMOS process are summarized in Table 2, which indicates that the overall performance of the designed circuit is robust against process/voltage/temperature (PVT) variations. Note that these results obtained from circuit simulations are in the typical NMOS typical PMOS (TT), slow NMOS slow PMOS (SS), and fast NMOS fast PMOS (FF) corner cases.

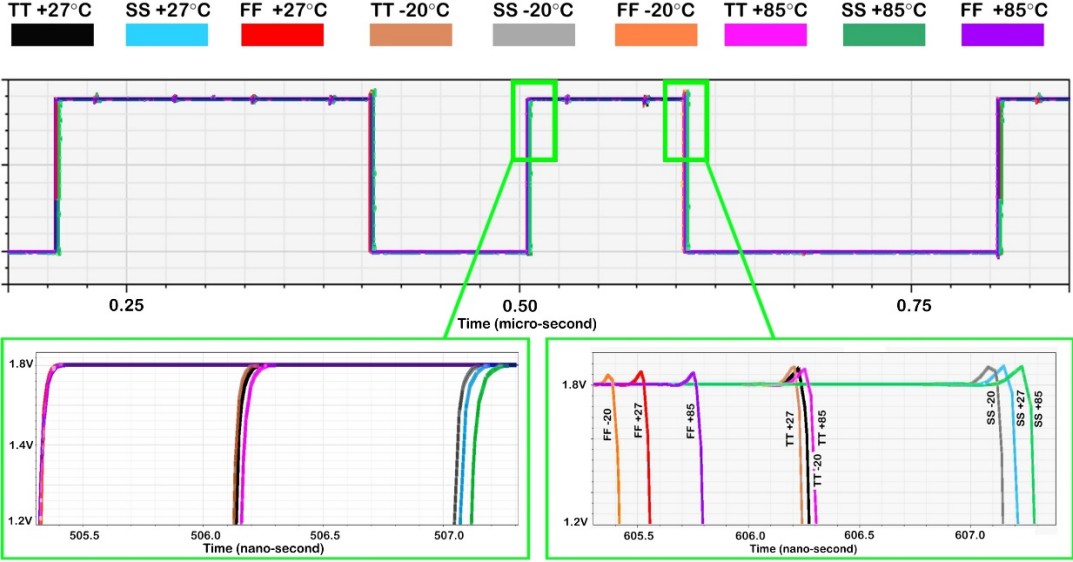

**Figure 8.** Process corners/temperatures simulation for demodulated data.

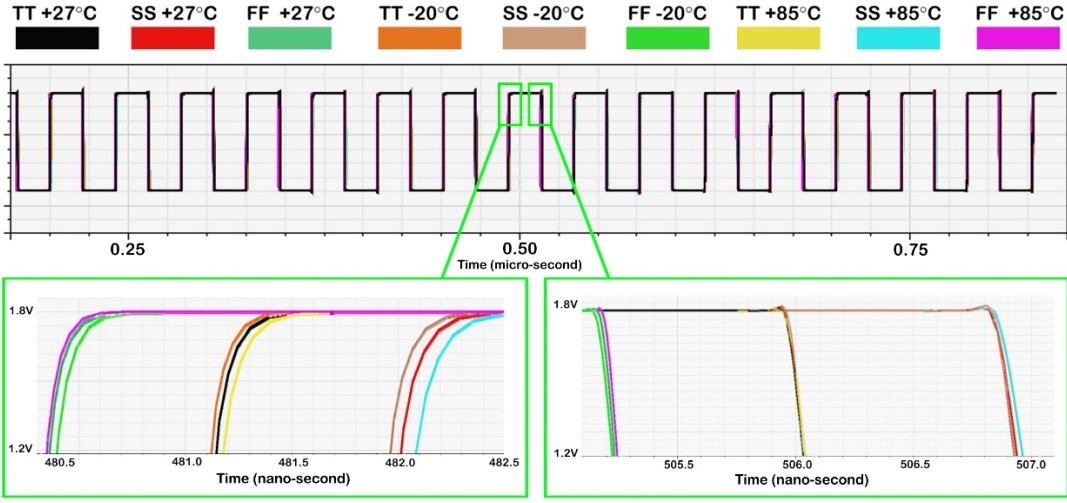

**Figure 9.** Process corners/temperatures simulation for the recovered clock.

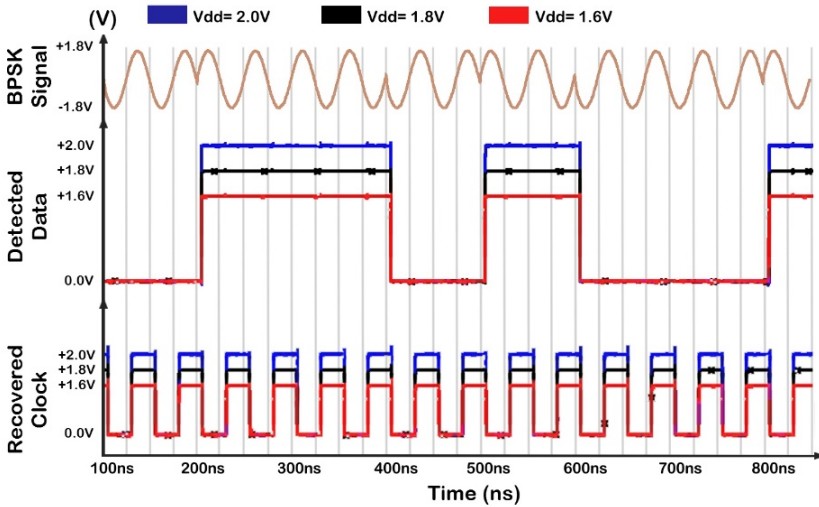

**Figure 10.** Demodulated data and the recovered clock with ±10% $V_{dd}$ variation.

**Table 2.** Performance summary of the proposed demodulator.

| Process Corner/Temperature | TT/27 °C | SS/85 °C | FF/−40 °C |
| --- | --- | --- | --- |
| Power Consumption * (1.8 V $V_{dd}$) | 8.7 μW | 5.9 μW | 13.2 μW |

* @ 20 MHz carrier frequency and 20 Mbps data rate.

As stated in Section 2, there is no need to place automatic gain control in the receiver in fixed-in position implantable biomedical applications where the minimum required level of received signal for correct operation of the circuit is provided by the power amplifier in the external part according to the link properties. However, in order to show the correct operation of the proposed circuit against variations on received BPSK signal level, Figure 11 depicts the post-layout-simulated behavior of the proposed demodulator with ±30% changes in the received BPSK signal level, for all of which correct operation was seen.

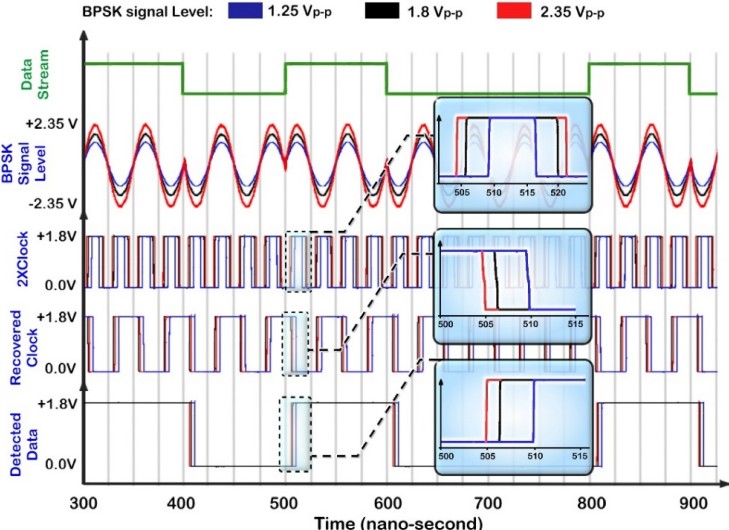

**Figure 11.** Demodulated data and the recovered clock with ±30% variations on received BPSK signal level.

In order to achieve accurate simulations considering fabrication process variations, the Monte Carlo analysis is performed on the designed BPSK demodulator by applying both process variations (such as variations on threshold voltage) and transistors' aspect ratio mismatches for all the devices

employed. Figures 12 and 13 show the simulated transient response of the proposed circuit for demodulated data and the recovered clock, respectively, with 500 iterations. As it can be seen, the effect of non-idealities on the performance of the proposed circuit is negligible. Moreover, Figure 14 illustrates the Monte Carlo simulation of power consumption with 100 iterations. As it can be seen, the circuit consumes as low as 8.5 μW from the power supply of 1.8 V at the data rate of 20 Mbps, and the highest deviation of power consumption from its typical value is almost 30%.

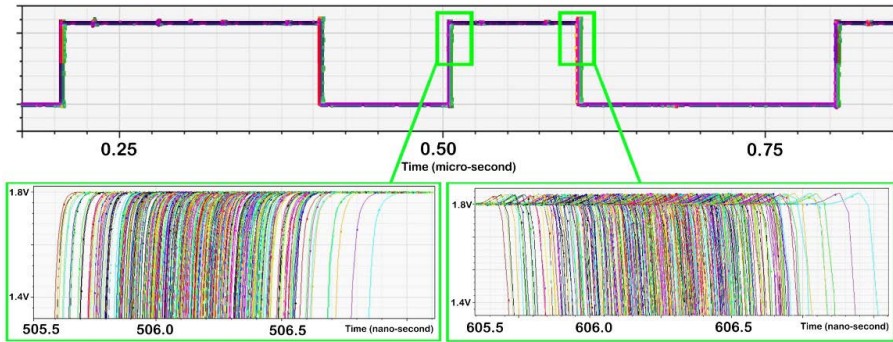

**Figure 12.** Monte Carlo simulations of demodulated data with 500 iterations.

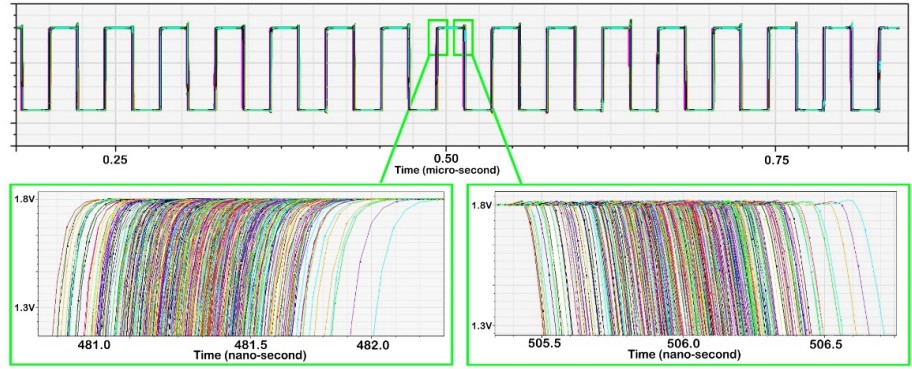

**Figure 13.** Monte Carlo simulation of the recovered clock with 500 iterations.

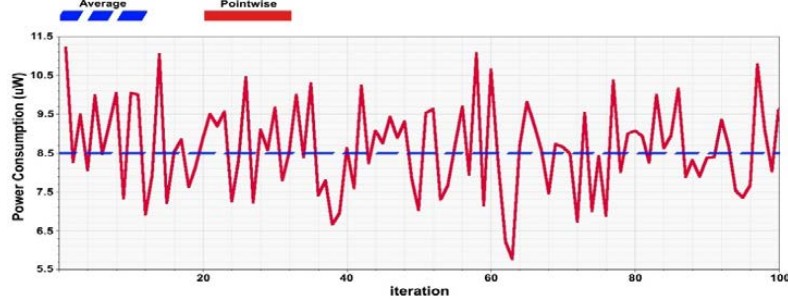

**Figure 14.** Monte Carlo simulation of power consumption with 100 iterations.

For verifying the functionality of the presented scheme in the circuit level, a proof-of-concept prototype was developed via TTL logic gates, a D flip-flop, an inductive coupling coil, and a diode rectification bridge, a photograph of which is shown in Figure 15. In this case, a 1 MHz, 1 Mbps BPSK signal with an amplitude of 5 V was employed as the input signal. Figure 16 shows an oscilloscope screen picture, presenting the important waveforms, which supports the proposed idea.

Performance comparisons between the proposed BPSK demodulator and other reported designs in the 0.18 μm CMOS process working at 1.8 V power supply are presented in Table 3. Note that the presented circuit is also simulated in different frequencies for making better comparisons.

The simulation shows that static power dissipation is dominant at low operating frequencies. It can be seen that the designed circuit works with extremely low power consumption and within a small occupied active area, with a DRCF ratio of as high as 100%.

**Table 3.** Performance comparisons with other similar works.

| | [7] | [8] | [10] | [11] | [14] | [15] | [16] * | [22] | [23] | [24] | This Work | | |
|---|---|---|---|---|---|---|---|---|---|---|---|---|---|
| Frequency (MHz) | 10 | 13.56 | 10 | 16 | 4 | 10 | 1 | 8 | 10 | 10 | 2 | 10 | 20 |
| Data rate (Mbps) | 10 | 1.12 | 10 | 16 | 0.8 | 10 | 1 | 8 | 10 | 10 | 2 | 10 | 20 |
| DRCF (%) | 100 | 8.25 | 100 | 100 | 20 | 100 | 100 | 100 | 100 | 100 | 100 | 100 | 100 |
| Chip Area ($\mu$m$^2$) | - | $19 \times 10^4$ | - | - | 4300 | 3500 | - | - | - | 1120 | 459 | 459 | 459 |
| Power Consumption ($\mu$W) | 119 | 610 | 232 | 27 | 59 | 77.9 | 88.2 | 148 | 27.2 | 14 | 5.6 | 6.7 | 8.5 |
| FOM $\left(\left(\frac{DRCF}{Power(\mu W)}\right) \times 10^{-3}\right)$ | 8.4 | 0.14 | 4.3 | 37 | 3.4 | 12.8 | 11.3 | 6.7 | 36.7 | 71.4 | 160 | 144 | 113 |

\* Without clock recovery circuit.

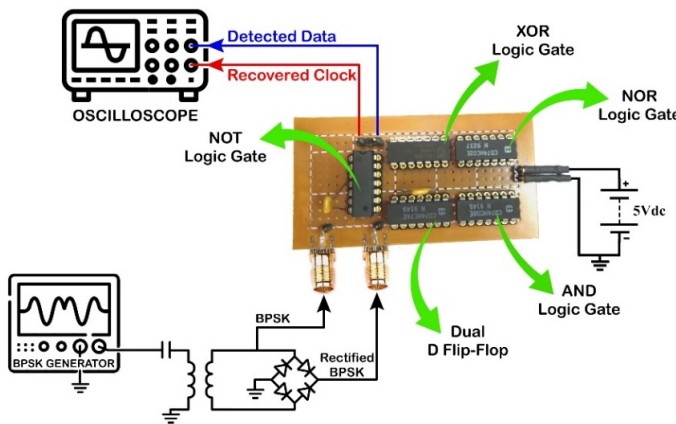

**Figure 15.** The tested proof-of-concept prototype.

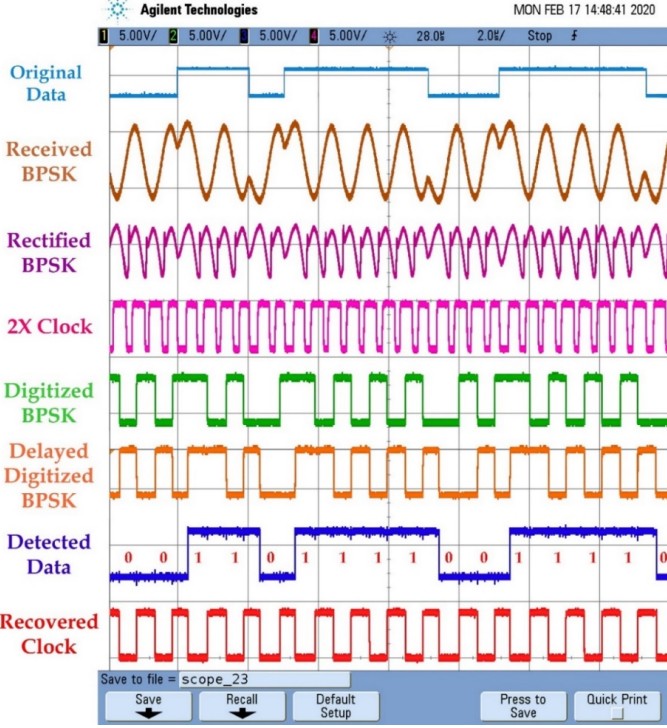

**Figure 16.** Experimental waveforms that resulted from the proof-of-concept prototype.

## 4. Conclusions

In this article, a low-power, high-data-rate, area-efficient BPSK demodulator utilizing a delayed digitized format of the input BPSK signal is presented. The presented circuit is designed and simulated in a standard 0.18 μm CMOS process using a 1.8 V power supply. Post-layout simulation and Monte Carlo analysis show that the presented circuit (including the clock recovery block) consumes 5.6 μW, 6.7 μW, and 8.5 μW at frequencies of 2 MHz, 10 MHz, and 20 MHz, respectively. The occupied active area of the whole circuit is $17 \times 27$ μm$^2$ and besides simplicity and low power consumption, the designed demodulator circuit benefits from a DRCF of 100%.

**Author Contributions:** Conceptualization and design, M.G.; formal analysis, M.G.; software, M.G.; investigation, M.H.M.; writing—original draft preparation, M.G.; writing—review and editing, M.H.M.; supervision, M.H.M., P.A., and S.H.-H.; funding acquisition, S.H.-H. All authors have read and agreed to the published version of the manuscript.

**Funding:** This research received no external funding.

**Conflicts of Interest:** The authors declare no conflict of interest.

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
