# Peer review of "An Ultra-Low-Power Area-Efficient Non-Coherent Binary Phase-Shift Keying Demodulator for Implantable Biomedical Microsystems"

_electronics, doi:10.3390/electronics9071123_

Round 1
Reviewer 1 Report
The acronyms should be specified the first time they are used; for example, “BPSK” is Binary Phase Shift Keying, it is trivial for me, but this might not be obvious for all potential readers. Others: PVT, …
I suppose an Automatic Gain Control (AGC) is needed, if the level of the signal PSK received is too low this receiver does not work. In such case, the 2xClock signal is zero permanently (ground); even the output of the rectifier can be zero. The 2xClock signal could be zero if the input if too low, this is because the “2xClock Extractor” is a digital buffer, two inverters CMOS in cascade connection.
Besides, if the input is too high, the duty cycle of 2xClock signal could be near to 100%. Authors suppose an input level that produces a 50% duty cycle in 2xClock signal. The effect of this duty cycle could be analysed in the receiver.
In Figure 7 authors must show the data in the modulator and the rectifier output. In Figure 7 authors suppose a peak value of 1.7 volts (approximately), this is a convenient value, I suppose the double of the threshold level for the inverter CMOS.
Authors study the process, voltage and temperature (PVT) variation effects, but they avoid study the effect of the received level of PSK signal in the receiver.
The meaning of TT, SS and FF should be specified for the process corner in Table 2.
The performance of the proposed receiver depends on the received level, for this reason when authors implement the TTL circuit they suppose a level of 5 volts in the PSK signal. This peak value produces good commutations; if the level were less than 1.2 (for example 0.5) the receiver does not work.
In Figure 15 authors must show the data in the modulator.
The proposed BPSK demodulator is similar to the implemented in reference 21. The main different is the circuit for generating the 2xClock signal, authors use a rectifier and a CMOS buffer; in 21 authors use a CMOS clipper, but both systems basically use the same signals generation; these implementations, seeing as block diagrams are similar. In other words, there are only few hardware implementations differences.
The proposed demodulator in reference 21 occupies more area, for the same CMOS technology, but meanly the reason is the free area in this layout.
It is true that this design uses less power than reference 21, but this is irrelevant since in both cases the effect of the input level is not analysed.
Author Response
Reply Letter
The authors would like to express their utmost gratitude to the anonymous reviewers for their thorough review and their valuable comments. Their comments have definitely helped improve the quality of the manuscript and also clarify some of the issues they have raised. Authors’ response to the reviewers’ comments are given below in blue. Moreover, the changes in the manuscript according to the reviewers’ points appear (in the revised manuscript) in blue. Taking the reviewers’ comments into consideration and based on editorial evaluation, some modifications are made on the previously submitted manuscript. Please note that all the figure numbers and reference numbers presented in the authors’ comments refer to the revised manuscript unless otherwise stated.
Reviewer Comments
Reviewer 1:
1- The acronyms should be specified the first time they are used; for example, “BPSK” is Binary Phase Shift Keying, it is trivial for me, but this might not be obvious for all potential readers. Others: PVT, …
Reply: We would like to thank the reviewer for the comment. The suggested corrections are edited and highlighted in the revised version of the manuscript.
2- I suppose an Automatic Gain Control (AGC) is needed, if the level of the signal PSK received is too low this receiver does not work. In such case, the 2xClock signal is zero permanently (ground); even the output of the rectifier can be zero. The 2xClock signal could be zero if the input if too low, this is because the “2xClock Extractor” is a digital buffer, two inverters CMOS in cascade connection.
Reply: We would like to thank the reviewer for its comment. The authors are well aware of the true fact that reviewer is pointing out. The performance of not only the proposed circuit, but most of the similar works presented in the literature depend on the received level. Placing an automatic gain control as the first stage of the receiver is needed in the works where orientation and matching between the transmitting and receiving blocks is lost, like in applications of transmitting data or power to freely-moving animals [17]. But, please note that as in fixed-in position implantable biomedical applications such as cochlear implants and visual prostheses [1] where the link properties such as distance between receiver and transmitter, orientation, and alignment are easily available, it is not needed to use automatic gain control in the receiver. In these cases, the minimum required level of received signal for correct operation of the circuit is provided by the power amplifier in the external part according to the link properties. Moreover, please note that in these cases, as the received BPSK signal level is high enough, power can also easily deliver wirelessly to the implant through the energy contained in the incoming BPSK data signal [18, 19]. Please note that some sentences are now added to the revised manuscript to discuss that it is not needed to use automatic gain control in the proposed circuit. We would like to draw the reviewer’s attention to the highlighted text in blue in the last paragraph of Section 1 and 1st paragraph of Section 2.
Besides the explanation provided above, to satisfy the reviewer, the proposed circuit was post-layout simulated with ±30% changes in the received BPSK signal level to show the tolerance level of the receiver against received signal level (See Fig. 11 of revised manuscript).
3- Besides, if the input is too high, the duty cycle of 2xClock signal could be near to 100%. Authors suppose an input level that produces a 50% duty cycle in 2xClock signal. The effect of this duty cycle could be analysed in the receiver.
Reply: We would like to thank the reviewer for its comment. The authors are in complete agreement with the issue that for very low and also high levels for the received signal the whole circuit may not work properly. Please note that as discussed in Q. 2, in the fixed-in position applications the required signal level is provided by the power amplifier in the external part according to the link properties. Consequently, there is no serious concern about the issue that reviewer is pointing out. Besides this explanation and to satisfy the reviewer, the proposed circuit was post-layout simulated with ±30% changes in the received BPSK signal level to show the tolerance level of the receiver against received signal level and accordingly changes in the duty cycle of 2XClock (See Fig. 11 of revised manuscript).
4- In Figure 7 authors must show the data in the modulator and the rectifier output. In Figure 7 authors suppose a peak value of 1.7 volts (approximately), this is a convenient value, I suppose the double of the threshold level for the inverter CMOS.
Reply: The authors are grateful for the reviewer for the comment. When doing circuit simulations, it may just need to show the original data in the modulator, which in now shown in Fig. 7 of the revised manuscript. Please note that as discussed in Q. 2 above, there is no concern regarding the received signal level in the fixed-in position applications, which is the case here.
5- Authors study the process, voltage and temperature (PVT) variation effects, but they avoid study the effect of the received level of PSK signal in the receiver.
Reply: The authors are grateful for the reviewer for the comment. Complete response to this question are given in Q. 2. Besides the explanation provided above, to satisfy the reviewer, the proposed circuit was post-layout simulated with ±30% changes in the received BPSK signal level to show the tolerance level of the receiver against received signal level (See Fig. 11 of revised manuscript).
6- The meaning of TT, SS and FF should be specified for the process corner in Table 2.
Reply: The authors would like to thank the reviewer for raising this issue. To incorporate the true concern of the reviewer, some sentences are now added to the revised manuscript. We would like to draw the reviewer’s attention to the 2nd paragraph of Section 3, where it says:
“… Note that these results obtained from circuit simulations in the typical NMOS typical PMOS (TT), slow NMOS slow PMOS (SS), and fast NMOS fast PMOS (FF) corner cases.”
7- The performance of the proposed receiver depends on the received level, for this reason when authors implement the TTL circuit they suppose a level of 5 volts in the PSK signal. This peak value produces good commutations; if the level were less than 1.2 (for example 0.5) the receiver does not work.
Reply: We would like to thank the reviewer for the comment. In general, the performance of the proposed receiver as well as other similar works reported in literature depend on the received signal level (that’s why AGC is needed in the receiver block in the applications where the link properties are not fixed which are not the case in this work) and the authors are well aware of the true fact that the reviewer is pointing out. Please note that as already stated in the text, Figs. 15 and 16 (of the revised manuscript) are just experimental results obtained from the proof-of-concept prototype. In fact, the proposed idea was just verified using off-the-shelf components (like the similar works reported in [15] and [23]) and if we had a chance to fabricate our IC, it was possible to show the operation of the proposed circuit for different received signal levels.
8- In Figure 15 authors must show the data in the modulator.
Reply: The authors are grateful for the reviewer for the comment. Taking into consideration the true concern of the reviewer, the original data in the secondary coil of isolating transformer is now shown in Fig. 16 of the revised manuscript.
9- The proposed BPSK demodulator is similar to the implemented in reference 21. The main different is the circuit for generating the 2xClock signal, authors use a rectifier and a CMOS buffer; in 21 authors use a CMOS clipper, but both systems basically use the same signals generation; these implementations, seeing as block diagrams are similar. In other words, there are only few hardware implementations differences. The proposed demodulator in reference 21 occupies more area, for the same CMOS technology, but meanly the reason is the free area in this layout. It is true that this design uses less power than reference 21, but this is irrelevant since in both cases the effect of the input level is not analysed.
Reply: We admit that the concept is not novel, but the authors would like to take the opportunity to provide a brief overview of the differences of this manuscript:
- In reference 21 (which is now reference 24 in the revised manuscript), some digital pulses are generated using the input BPSK signal by the CMOS clipper circuit, which is then utilized for final data detection. However, in the circuit presented in this paper, the data detection operation is performed using the rectifier circuit inside the power management block without the need for auxiliary circuits, which reduces the overall power consumption and occupied area of the circuit.
- In reference 21 (which is now reference 24 in the revised manuscript), to recover the CLK signal which is then used for data detection, a frequency divider is used. Since the output value for this circuit is not known at start-up time, it is very likely that the CLK signal will reverse each time the signal is applied to the circuit. This also reverses the output data. While in the presented work, a data detector core is proposed by which the BPSK signal is detected with the minimum possible error.
- Please note that as he reviewer mentioned above, there is some free area in the layout of [24], but if we redraw the layout in more compact way, it would result in maximum 25% saving area which is still twice the circuit area of the presented work.
Reviewer 2 Report
Well written paper. The quality of the figures could be improved. As an example, in Fig. 7, 12, please change the X-axis unit to micro-second instead of 'us'. It appears that the authors have not considered any threshold voltage variations in the transistors. How have the authors considered the impact of process variations in their PVT analysis? Can the authors comment on the reason for not considering that?
Author Response
Reply Letter
The authors would like to express their utmost gratitude to the anonymous reviewers for their thorough review and their valuable comments. Their comments have definitely helped improve the quality of the manuscript and also clarify some of the issues they have raised. Authors’ response to the reviewers’ comments are given below in blue. Moreover, the changes in the manuscript according to the reviewers’ points appear (in the revised manuscript) in blue. Taking the reviewers’ comments into consideration and based on editorial evaluation, some modifications are made on the previously submitted manuscript. Please note that all the figure numbers and reference numbers presented in the authors’ comments refer to the revised manuscript unless otherwise stated.
Reviewer Comments
Reviewer 2:
1- Well written paper. The quality of the figures could be improved. As an example, in Fig. 7, 12, please change the X-axis unit to micro-second instead of 'us'.
Reply: The authors are pleased to hear this, and would like to thank the reviewer for his/her kind statement. The suggested corrections are edited in the revised version of the manuscript. Moreover, we have tried to improve the quality of figures. Please note that one figure is now added to the revised version of the manuscript and consequently Fig. 12 is renamed to Fig. 13 in the revised manuscript.
2- It appears that the authors have not considered any threshold voltage variations in the transistors. How have the authors considered the impact of process variations in their PVT analysis? Can the authors comment on the reason for not considering that?
Reply: The authors would like to thank the reviewer for the comment. This important issue was already considered in the previous version of the manuscript. But, to incorporate the true concern of the reviewer, some sentences are now added to the revised manuscript. We would like to draw the reviewer’s attention to the 3rd paragraph of Section 3, where it says:
“In order to achieve accurate simulations considering fabrication process variations, the Monte Carlo analysis is performed on the designed BPSK demodulator by applying both process variations (such as variations on threshold voltage) and transistors’ aspect ratios mismatches for all the devices employed. …”
It should be added that in the submitted manuscript, process variations were studied using default settings of a standard 0.18-µm CMOS technology via Monte Carlo simulations in Cadence (Fig. 12 to 14) and the behavior of the circuit was simulated over the rather wide temperature range of -20o to +85o (Fig. 8 and Fig. 9) and also 10% changes in power supply (Fig. 10) for all of which correct operation was seen.
Round 2
Reviewer 1 Report
Dear authors,
thank you for your answer. I now understand the situation regarding with the level of the received signal.
Best regards.